# A comprehensive antigen-antibody complex database unlocking insights into interaction interface

Yuwei Zhou[1], Wenwen Liu[1], Ziru Huang[1], Yushu Gou[1,2], Siqi Liu[1], Lixu Jiang[1], Yue Yang[1], Jian Huang[1,2]*

[1]The Clinical Hospital of Chengdu Brain Science Institute, School of Life Science and Technology, University of Electronic Science and Technology of China, Chengdu, China; [2]School of Healthcare Technology, Chengdu Neusoft University, Chengdu, China

*For correspondence:
hj@uestc.edu.cn

## eLife Assessment

This **useful** manuscript provides a newly curated database (termed AACDB) of antibody-antigens structural information, alongside annotations that are either taken and from the PDB, or added de-novo. Sequences, structures, and annotations can be easily downloaded from the AACDB website, speeding up the development of structure-based algorithms and analysis pipelines to characterize antibody-antigen interactions. The methodology presented for this data curation is **solid**. The curated dataset will be of broad interest and value to researchers interested in antibody-antigen interactions.

**Abstract** Antibodies are critical components of the vertebrate immune system and possess a wide array of biomedical applications. Elucidating the complex interactions between antibodies and antigens is an important step in drug development. However, the complex and vast nature of the data presents significant challenges in accurately identifying and comprehending these interactions. To overcome these challenges and deepen our understanding of the antibody-antigen interface, we developed the Antigen-Antibody Complex Database (AACDB). The current version provides a comprehensive collection of 7498 manually processed antigen-antibody complexes, ensuring accuracy and detail. This database provides extensive metadata and rectifies annotation errors found in the PDB database. Furthermore, it integrates data on antibody developability and antigen-drug target relationships, making it valuable for assisting new antibody therapies development. Notably, the database includes comprehensive paratope and epitope annotation information, thereby serving as a valuable benchmark for immunoinformatics research. The AACDB interface is designed to be user-friendly, providing researchers with powerful search and visualization tools that enable effortless querying, manipulation, and visualization of complex data. Researchers can access AACDB completely online at http://i.uestc.edu.cn/AACDB. Regular updates are promised to ensure the timely provision of scientific and valuable information.

## Introduction

Antibodies are not only an essential part of the vertebrate immune system, but also have wide application in biomedical field (*Lyu et al., 2022*; *Su and Shuai, 2020*). In clinical practice, the advent of monoclonal antibodies (mAbs) has transformed the treatment for various cancers, autoimmune diseases, and numerous other conditions. Due to their large binding interfaces, high affinity and specificity,

antibody drugs have successfully targeted numerous proteins that were previously impervious to small molecule therapies. The expansion of antibody drug formats, such as antibody-drug conjugates (ADCs), domain antibodies, bispecific antibodies, and antibody fusion proteins, has broadened their indications to a wider range of diseases including blood disorders, infections, neurological conditions, ocular diseases, and metabolic disorders (*Carter and Rajpal, 2022*; *Tsuchikama et al., 2024*; *Chi et al., 2023*). Antibody drug development typically encompasses three stages: three stages: preclinical research, clinical trial, and post-marketing surveillance study (*Zhou et al., 2023*). The preclinical research of antibodies mainly refers to a series of in vitro experiments and animal model studies conducted before the application of antibodies in clinical trials. In this stage, an important task is to determine the binding interfaces and interaction sites between antibodies and their drug targets. Understanding how antibodies specifically interact with their targets or antigens can help optimize antibody drug leads further, as well as provide insights into mechanisms of interaction (*Myung et al., 2023*).

The field of machine learning and deep learning-based methods for antigen and antibody binding interfaces prediction has gained significant attention in recent years (*Pittala and Bailey-Kellogg, 2020*; *Qiu et al., 2023*; *Zhou et al., 2019*; *Qi et al., 2014*; *Papadopoulos et al., 2025*; *Chinery et al., 2023*; *Li et al., 2024b*; *Liberis et al., 2018*; *Kalemati et al., 2024*; *Shashkova et al., 2022*). The advancement of this field heavily relies on a reliable foundation of data. Although the Protein Data Bank (PDB) (*Burley et al., 2023*; *Burley et al., 2024*; *Berman et al., 2000*) is a well-established repository for protein complexes, identifying antigen-antibody complexes within its extensive collection of general protein structures presents considerable challenges. In 2018, AbDb (*Ferdous and Martin, 2018*) was introduced as a specialized database for antigen-antibody complexes. However, its adoption has been limited by the absence of advanced visualization tools, interactive functionalities, and abundant metadata annotations. We acknowledge that AbDb has made exceptional contributions to the field by aggregating antibody-antigen structures over the past decade. Our work builds upon these foundations, aiming to address complementary challenges in annotation and interoperability. In comparison, the series of SAbDab databases (*Dunbar et al., 2014*; *Raybould et al., 2020*; *Schneider et al., 2022*) indeed provide a comprehensive data source of antibody structure and achieve in time updating. However, this rapid updating process may inadvertently overlook a significant amount of information that requires thorough verification, thereby increasing the likelihood of incorporating erroneous annotations directly from the PDB database.

A further challenge in the field of interface prediction is the lack of a standardized definition for interacting amino acids. Some datasets are based on differences in the solvent accessible surface area (ΔSASA) for each residue upon binding (*Qiu et al., 2023*; *Sun et al., 2009*); others are based on distances between atoms (*Pittala and Bailey-Kellogg, 2020*; *Chinery et al., 2023*; *Li et al., 2024b*; *Liberis et al., 2018*; *Daberdaku and Ferrari, 2019*). Different studies employ different datasets and definitions, resulting in a lack of comparability. There is an urgent need for a standard benchmark dataset to enable objective comparisons of these methods. None of the databases mentioned above provide details of interaction residues. To address these limitations, we developed the Antigen-Antibody Complex Database (AACDB). It has a user-friendly interface for convenient querying, manipulation, browsing, and visualization of comprehensive information about antibody-antigen complexes. Notably, AACDB also provides detailed information on interacting residues using two methods (ΔSASA and atom distance), facilitating the benchmarking studies of the predictive methods for paratopes and epitopes.

## Results

### Statistics

From over 32,000 experimental structures in the PDB database, we manually curated 7498 antigen-antibody entries for the current version of AACDB, encompassing to 16 antibody fragment types across 14 species (*Figure 1*). It is obvious that Fab fragments and human antibodies accounted for the largest proportion of the data, accounting for 71.98% and 60.95%, respectively. Our statistical analysis reveals a significant increase in the number of antigen-antibody complex entries within the PDB during the period from 2021–2023. These entries accounted for approximately 45% of the total antigen-antibody complex entries. Our search was conducted with an end date of November 2023.

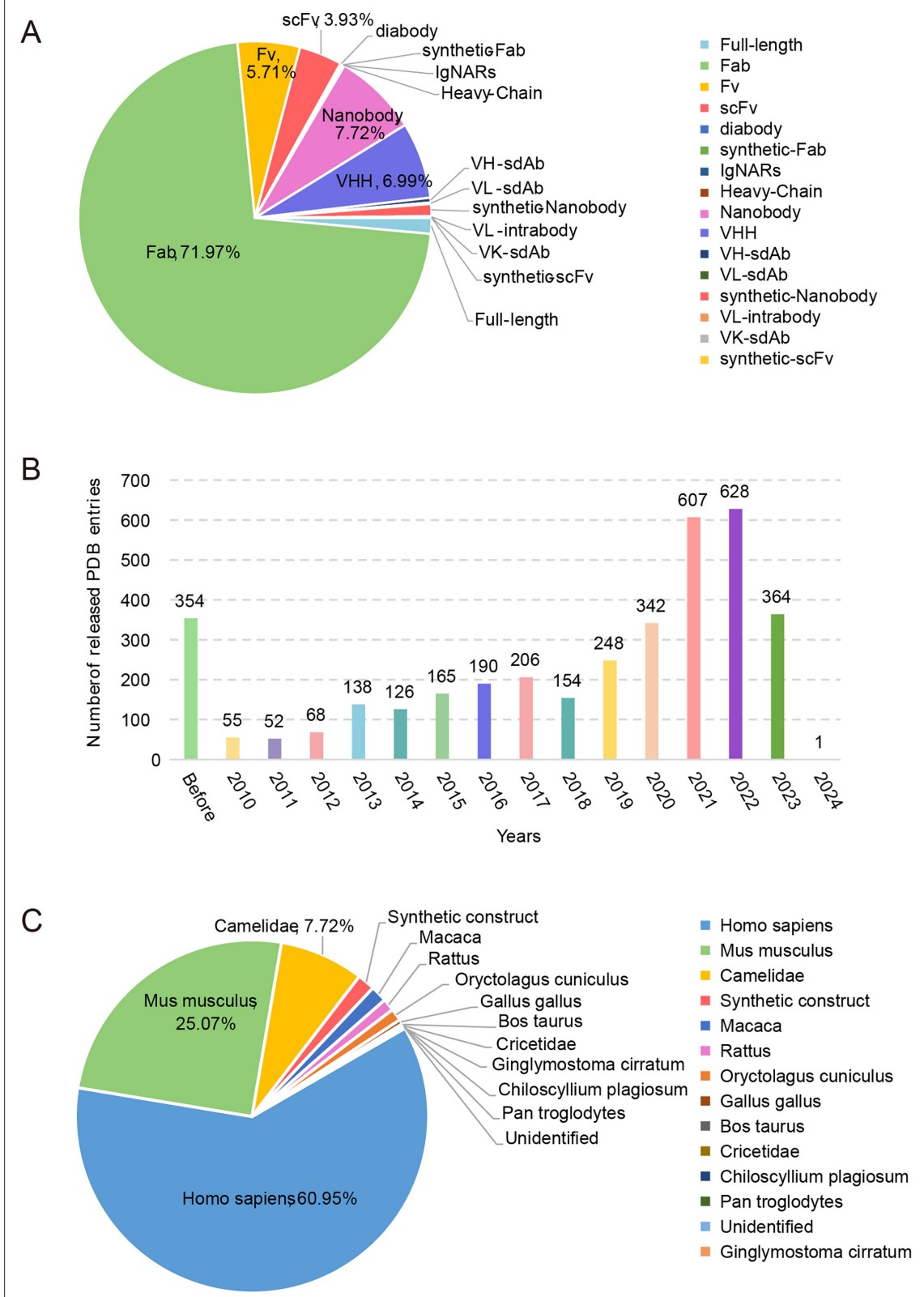

**Figure 1.** Antigen-Antibody Complex Database (AACDB) statistics. (**A**) Antibody fragment distribution in the database. (**B**) The number of antibody-antigen complexes released in different years (unique PDBID). (**C**) Organismal distribution of antibody entries. Statistical cutoff date: May 30, 2024. Fab: antigen binding fragment; Fv: variable fragment; scFv: single chain variable fragment; VHH: Variable domain of heavy chain of heavy chain antibody; sdAb: single domain antibody.

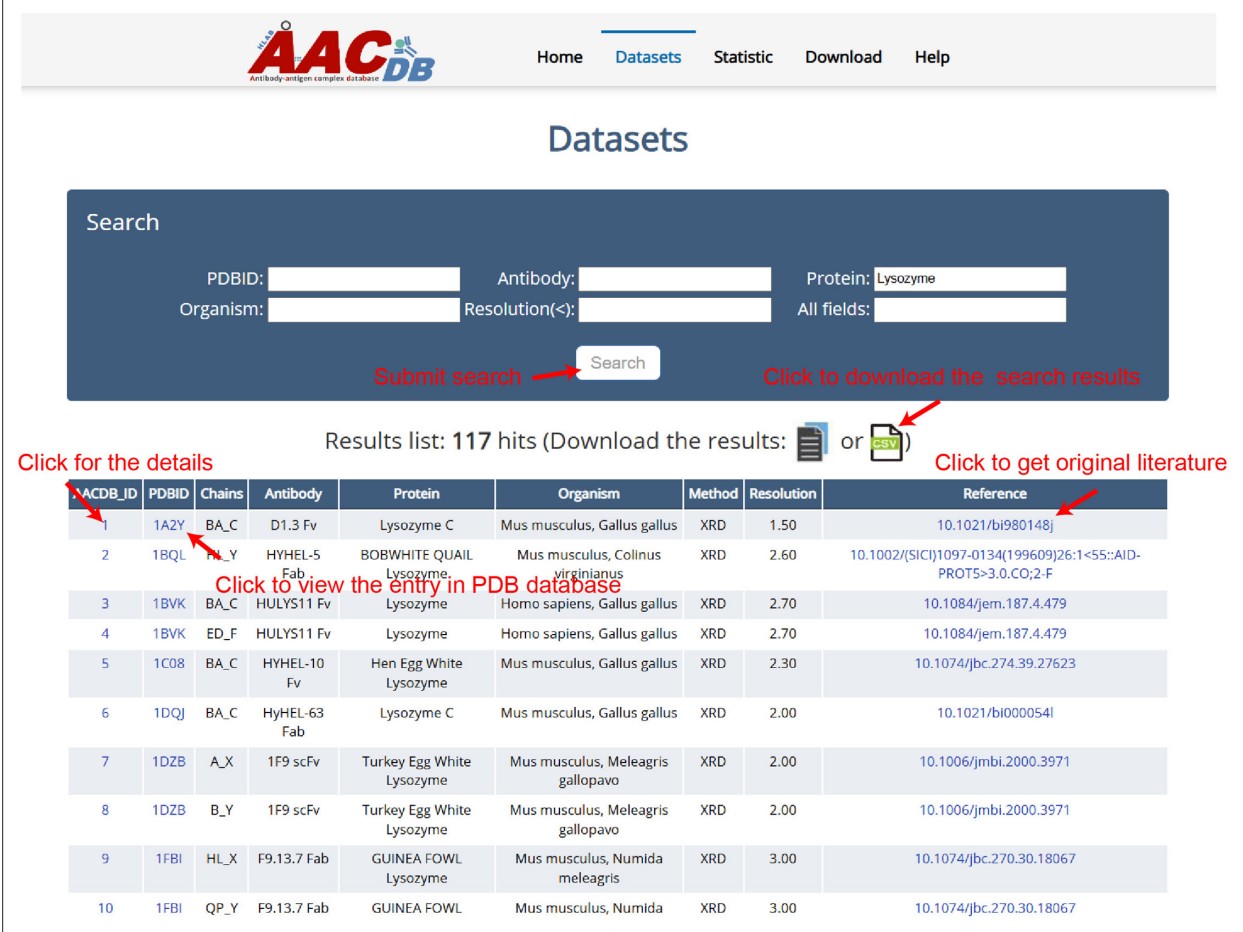

**Figure 2.** The Antigen-Antibody Complex Database (AACDB) browse and search page. This example demonstrates the use of the keyword 'Lysozyme' in the 'Protein' column, showcasing the search functionality of the AACDB database. The search results display an antigen related to lysozyme.

Notably, *Figure 1B* contains a 2024-dated entry resulting from database versioning updates: the original 7SIX record (added November 16, 2022) was superseded by the 8TM1 entry through an official revision on January 12, 2024. Furthermore, the developability properties of antibodies in 325 entries can be queried in the DOTAD database, at the meanwhile, 3733 antigen records have been identified as drug targets in DrugBank (data not shown).

## Database browse and search

All the data can be browsed directly by clicking the 'Datasets' item on the top menu of AACDB webpage (http://i.uestc.edu.cn/AACDB; *Figure 2*). The summary table includes nine columns as follows:

1. AACDB_ID: The unique id in the AACDB database, linking to the 'Detail' page;
2. PDBID: The identifier of the RCSB PDB database (https://www.rcsb.org/), linking to PDB;
3. Chains: The chain ids contained in this entry. Antibody and antigen chains were separated by '_.' The heavy chain id precedes the light chain id if antibody is complete;
4. Antibody: This column is represented by 'antibody name +fragment;'
5. Protein: The corresponding antigen;
6. Organism: The source organism of antibody and antigen;
7. Method: The experiment method used to solve the structure;
8. Resolution: The indicator that measures the resolution of protein structures in experiments is expressed in units of angstroms (Å).
9. Reference: The DOI linker of original literature that produced this structure.

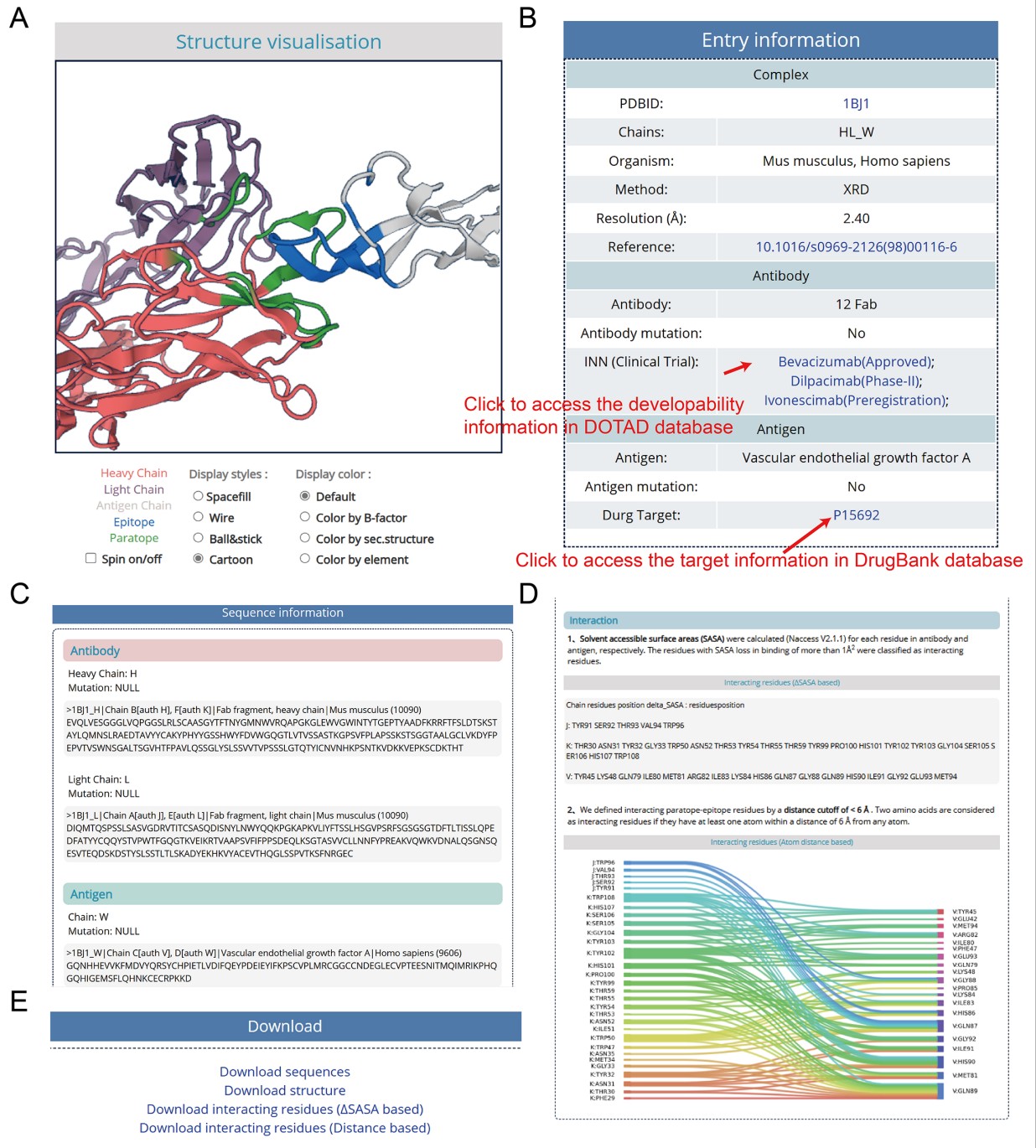

**Figure 3.** The details page of entries, taking '1BJ1' as an example. (**A**) Structure visualization window. The molecular structure is displayed with distinct color-coding for clarity: each chain is represented by a unique color, facilitating easy identification. The epitope is highlighted in blue, while the paratope is marked in green (**B**) Entry meta information. (**C**) Sequence and mutation information. (**D**) Interacting residues details based on solvent accessible surface area (SASA) and atom distance methods. (**E**) The download hyperlinks of a single entry.

The table in AACDB can be easily searched through the search panel at the top section of the 'Datasets' page. Users can perform a quick search by specifying one or more fields and entering relevant keywords. The search results can be downloaded as files in either txt or csv format. *Figure 2* shows the search results using the keyword 'Lysozyme' in 'Protein' column, returning 117 hits.

An individual structure can be accessed using its AACDB_ID accession code. When clicking the AACDB_ID hyperlink, the user will be brought to its details page as shown in *Figure 3*. There, the

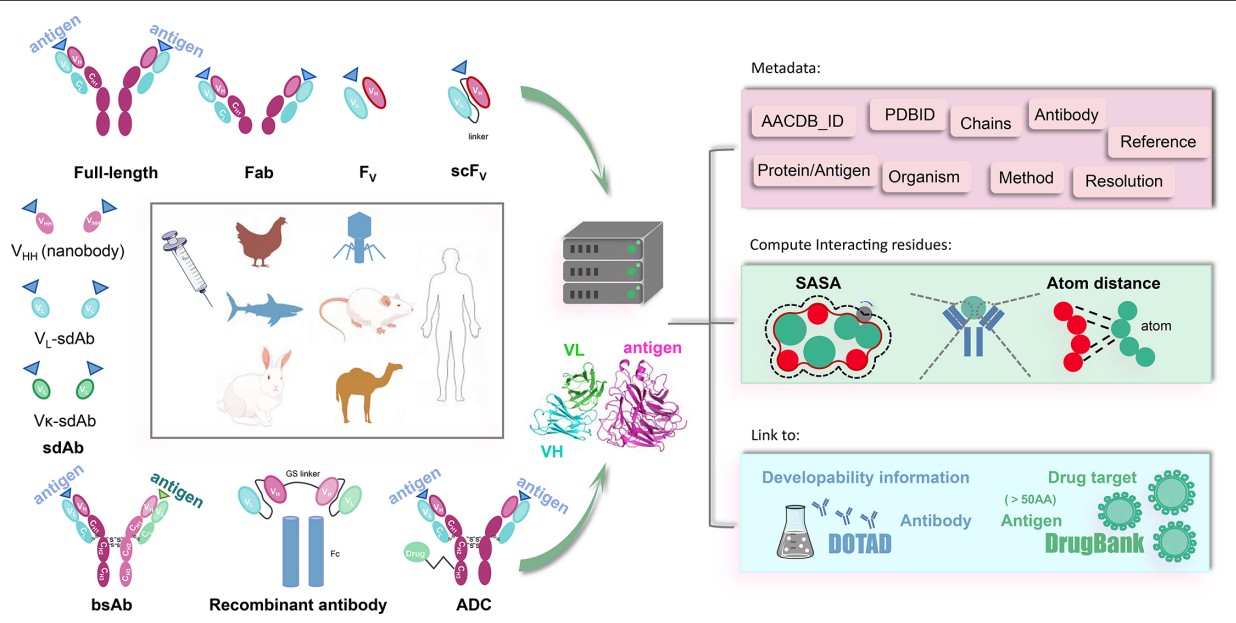

**Figure 4.** The Antigen-Antibody Complex Database (AACDB) framework integrates standardized metadata, interface annotation, and antigen-drug target relationships for comprehensive antigen-antibody complex analysis.

complex structure can be visualized with different colors and styles (*Figure 3A*). Besides the visualization window, we provide more details of this entry, including the mutation, INN, and clinical trial of antibody, the ID in DrugBank of antigen (if it exists) (*Figure 3B*). Under the structure information tab, further details about each chain can be found. These include: sequence, mutate amino acid type and position, interacting residues in each chain based on the ΔSASA method, interacting plot of paratope-epitope residues by a distance cutoff of <6 Å (*Figure 3D*). It should be pointed out that while our manuscript emphasizes widely accepted thresholds for consistency with prior benchmarks, AACDB explicitly provides raw ΔSASA and distance values for all annotated residues. Users can dynamically filter the data from the downloaded file (e.g. selecting 5 Å instead of 6 Å) by excluding entries exceeding their preferred thresholds.

## Data download

We provide two ways for downloading the data:

1. **Download data of the single entry:** When accessing the detailed information about an entry using the corresponding AACDB_ID, user can click the hyperlink at the 'Download' section of the bottom of the page to download the data for a single certain entry (*Figure 3E*).
2. **Download all the data of AACDB:** AACDB provides the download page for users. All the sequence and structure files and the interacting data based on different methods were packaged in different .zip files that can be downloaded.

Moreover, the website provides a user-friendly 'Help' page that presents a step-by-step tutorial to assist users in manipulating, querying, browsing, and downloading the AACDB database.

## Discussion

Research on antigen-antibody interactions contributes to the advancement of the antibody-related industry. Databases such as PDB and SabDab provide the foundational data for this purpose. However, there are still many unresolved issues. In this study, we developed the AACDB database to provide a curated and reliable dataset of antigen-antibody interactions. During the process of data collection and organization, we identified numerous annotation errors in the PDB database. Some of these errors had been directly introduced into SabDab. For example, the species of the antibody in 7WRL was incorrectly labeled as 'SARS coronavirus B012' in both PDB and SabDab. We also invested significant

effort and time to manually cross-reference with original literature in order to rectify these errors and exclude antibody binding proteins that were erroneously annotated as antigens by SabDab. Apart from the curation and reannotation of structural data, AACDB offers features not provided by other antigen-antibody complex databases: (1) AACDB's data processing pipeline supports mmCIF files, and (2) we provide amino acids in the interaction interface through two methods, enabling the definition of unified standards for epitopes and paratopes. This provides a more accurate and comprehensive benchmark dataset for developed interaction interface prediction tools, enhancing the comparability of various tools.

However, AACDB still has some limitations. Despite our best efforts, the limitations of our team's resources and knowledge mean that our database may not capture all antigen-antibody complex structures. Since the data is manually curated, eliminating errors completely during the information processing is a challenge. While we strive to fill in any gaps, we also hope that experts and users within the community can provide timely feedback to help us improve these issues. Additionally, currently, AACDB only includes antigen proteins with a length greater than 50 amino acids. The current AACDB release exclusively includes antigen proteins exceeding 50 amino acids in length. Future iterations will

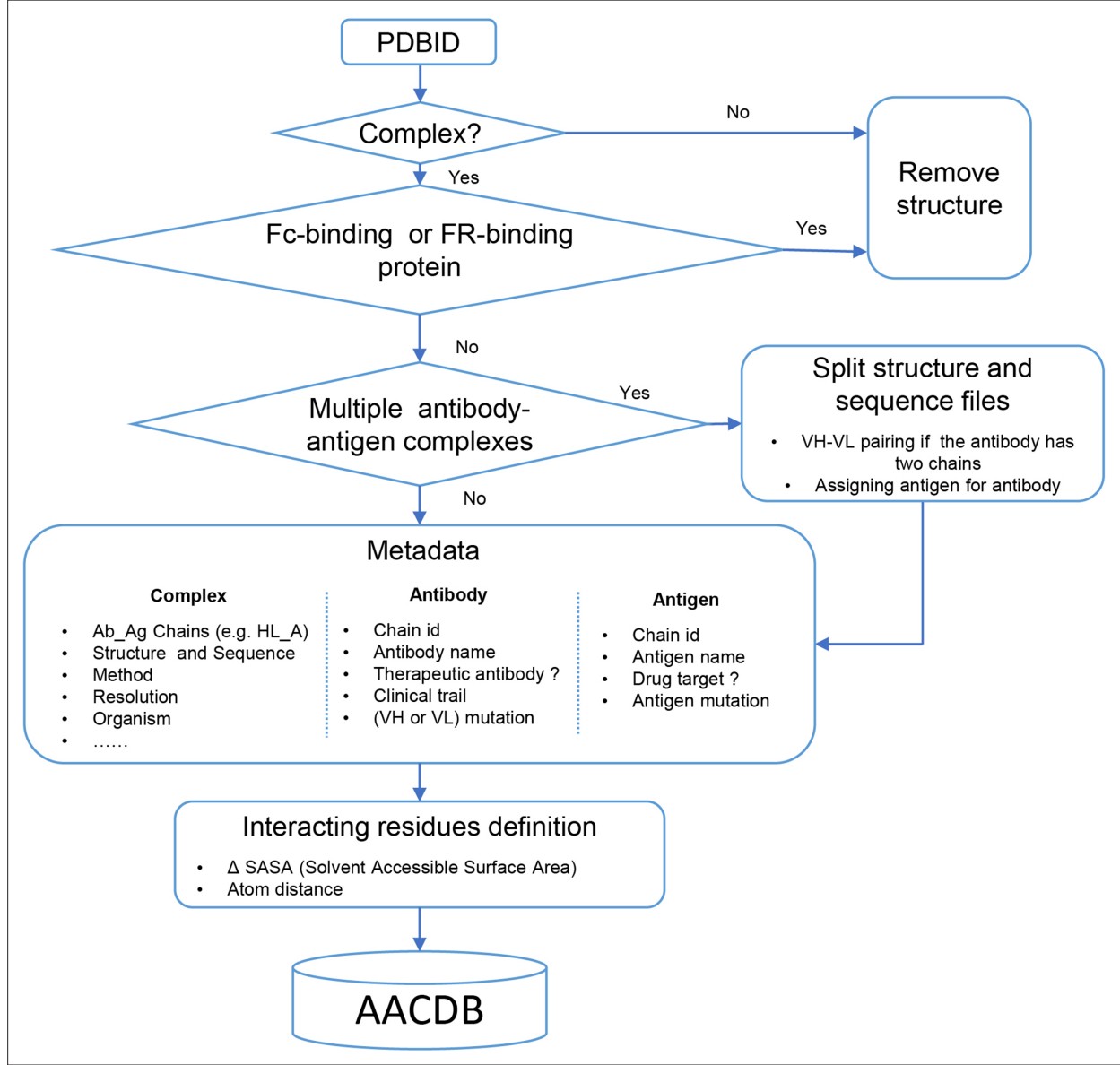

**Figure 5.** Data processing algorithm pipeline of Antigen-Antibody Complex Database (AACDB).

expand structural coverage to diverse antigen classes (peptides, nucleic acids, haptens) while incorporating literature-validated affinity data through systematic cross-referencing.

In summary, AACDB is a novel database of antigen-antibody complexes that provides information on antibody-developability, antigen-drug target relationships, and detailed antigen-antibody interaction interfaces. It is fully accessible at http://i.uestc.edu.cn/AACDB. We are committed to a regular data update, ensuring that researchers in immunoinformatics have access to timely and valuable resources.

## Materials and methods

The conceptual framework of AACDB is illustrated in *Figure 4*. Based on this concept, we first search the PDB database (https://www.rcsb.org/) using the keywords 'antibody,' 'antigen,' 'immunoglobulin,' and 'complex.' Over 32,000 experimental structures were extracted (up to December 2023). Each entry was next processed following the steps in *Figure 5*.

### Entry screening

We aimed to retrieve all antibody-related complexes from the PDB using the chosen search approach. However, we encountered a notable number of false hits in the results, prompting us to implement a stringent structural filtering process. Antibody complexes were defined as structures containing at least one antibody molecule and another protein exceeding 50 amino acids in length. Structures that lacked antibodies or consisted solely of one type of antibody were excluded from further analysis. It is crucial to emphasize that not all proteins that bind to antibodies qualify as antigens. For instance, immunoglobulin-binding proteins like Protein A and Protein G, expressed by *Staphylococcus aureus* and Streptococcal species, are commonly employed for antibody purification procedures (*Fishman and Berg, 2019*). Although protein A may bind to the Fab region of antibodies, the interacting amino acids might be located in the framework region (FR) rather than the complementarity-determining regions (CDRs) (*Graille et al., 2000*). These proteins are often misidentified as antigens. Consequently, structures exhibiting such characteristics require meticulous verification and should be excluded from our dataset.

### PDB splitting

We directly incorporated the PDB entry of antigen-antibody complex with only one antigen and one antibody into the AACDB database. However, quite a few antigen-antibody complexes contain several antigens and antibodies. In such cases, before splitting the structures, we need to determine the correct pairing of light and heavy chains by examining the information on equivalent chain interactions in the PDBsum database (*Laskowski et al., 2018*). Next, we assign the correct antigen chains to the antibodies by identifying whether the remaining chains in the structure interact with the CDR regions of determined antibodies. For example, in the case of 1AHW, which contains two copies of antigen-antibody complexes, chains BA and DE are identified as two antibody pairs, where chains C and F bind to BA and DE, respectively. Consequently, 1AHW is split into two files, BAC and DEF (*Figure 6A*). In 6OGE, Pertuzumab (chains C and B) and Trastuzumab (chains E and D) bind to different epitopes of the receptor tyrosine-protein kinase erbB-2 (chain A). This generates two records in AACDB (CBA and EDA) (*Figure 6B*). Specifically, in the analysis of anti-idiotypic antibody complexes, the partner antibody is treated as a dual-chain antigen, necessitating individual evaluation of heavy chain and light chain interactions with the anti-idiotypic component. When both chains engage in binding, the complex is divided into two distinct entries: 'anti-idiotypic antibody-heavy chain' and 'anti-idiotypic antibody-light chain.' Conversely, if only one chain participates, the non-interacting chain will be excluded from AACDB records. For instance, in 3BQU, the anti-idiotype 3H6 Fab (chains DC) only interacts with the heavy chain (chain B) of 2F5 Fab. In AACDB, 2F5 Fab will be split and the light chain (chain A) will be discarded (*Figure 6C*).

For each PDB entry, we utilized the corresponding split chains information to divide the downloaded.pdb based on the ATOM records. Notably, certain structures are exclusively provided in the 'mmCIF' format by the PDB database. While '.pdb' and '.cif' files store atomic coordinates in distinct text formats, the segmentation of these structure files is automatically performed based on manually annotated antibody-antigen chains. To address this, we integrated these considerations into our file

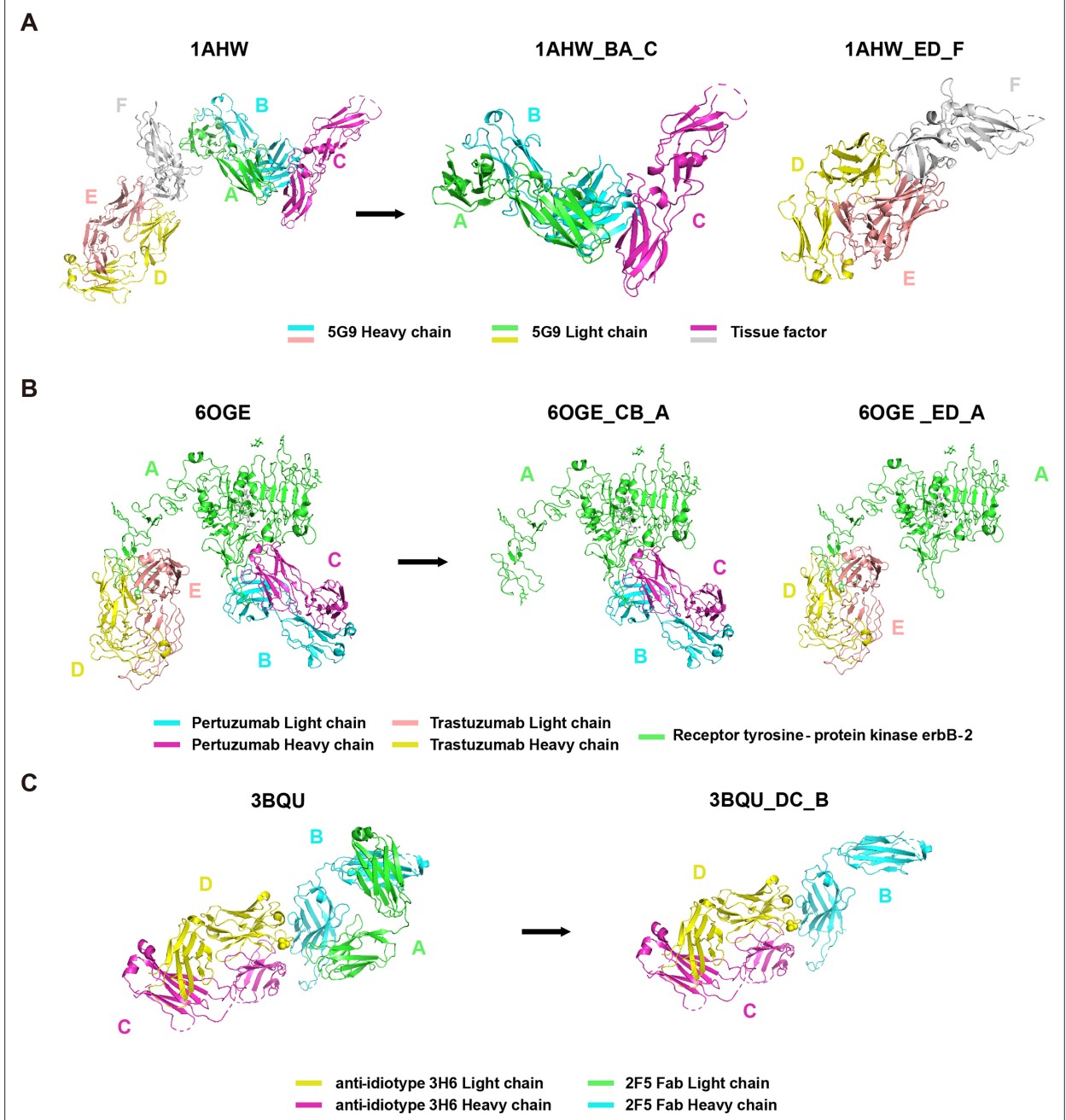

**Figure 6.** Examples of Protein Data Bank (PDB) file splitting under different situations. (**A**) 1AHW contains two copies of the same antigen and the same antibody. (**B**) In 6OGE, two different antibodies bind to distinct epitopes of the same antigen. (**C**) In 3BQU, an anti-idiotypic antibody binds exclusively to a single chain of the antibody.

processing pipeline enabling a fully automated file segmentation process. Additionally, the Naccess software does not support .cif files as input, we converted the '.cif' files into '.pdb' format files while performing the splitting process (**see Interacting residues definition**). Furthermore, we refined the annotation of the split '.fasta' files to ensure coherence with AACDB records.

## Metadata

AACDB provides detailed metadata for each entry, including chain IDs, antibody name, antigen name, method, resolution, organism, and more. To ensure data accuracy, we have conducted comprehensive verification by consulting original literature sources. AACDB has addressed many annotation errors

identified within the corresponding PDB entries. These errors include but not limited to: (1) mislabeling of species (e.g. the entry 7WRL where the organism of BD55-1239H antibody was erroneously labeled as 'SARS coronavirus B012'); (2) Resolution annotation errors (e.g. 1NSN, in which the resolution of 2.9 Å is misannotated as 2.8 Å).; (3) mislabeling of antibody chains as other proteins (e.g. in 3KS0, the light chain of B2B4 antibody was misnamed as heme domain of flavocytochrome b2); (4) misidentification of heavy chains as light chains (e.g. both two chains of antibody were labeled as light chain in 5EBW); (5) mutation status annotation errors. We have identified cases in which PDB entries indicate 'NO' for mutations, while in reality, mutations exist (e.g. bevacizumab (Avastin) in 6BFT was labeled as none mutation. When aligned with the bevacizumab sequence, however, mutation T8D/T30D in heavy chain and S52D/S53D in light chain were observed.); and (6) incomplete annotations. Certain entries only provide the name of the mutant without specifying the precise mutation site (e.g. in 7SU1, antibody was described as Ipilimumab variant Ipi.106. but PDB database does not provide any mutation amino acid or position, which can be identified by blast to Ipilimumab sequence).

We carefully checked each entry manually to find out all possible annotation problems. Given the context-dependent nature of mutations, sequence alignment was not universally performed across all antibody sequences. Instead, therapeutic antibody sequences served as definitive references. For example, for entries in the PDB labeled as 'Bevacizumab mutant' or 'Ipilimumab variant Ipi.106' where the 'Mutation(s)' field was annotated as 'NO,' we retrieved corresponding therapeutic antibody sequences from Thera-SAbDab and performed sequence alignment with the PDB entries to identify mutated residues. For other antibodies lacking well-defined wild-type references, mutation information was directly extracted from the PDB or original literature. All corrections have been publicly archived in AACDB.

The antibody nomenclature follows the title of the corresponding search entry in the RCSB PDB database, with verification done through the original literature. In cases where the name in the original literature differs from that in the RCSB PDB, we used the name in the published literature as the standard. Furthermore, for antibody fragments lacking names in both the RCSB PDB database and original literature, we adopt a naming convention of 'PDBID' + 'antibody fragment' (e.g. 4WEB Fab).

Biological and physicochemical properties are critical considerations in the development pipelines for therapeutic antibodies. These properties include solubility, immunogenicity, aggregation tendencies, expression level, stability, and hydrophobicity. We provide the International Non-proprietary Name (INN) and clinical trial information for each therapeutic antibody entry, linked to the DOTAD database (*Li et al., 2024a*). Numerous antigens have been successfully identified as targets for antibodies or small molecule drugs. We conducted a comparison between antigen sequences and the drug targets listed in the DrugBank database (*Knox et al., 2024*). A threshold of percent identity >90% was applied to determine the corresponding drug targets.

## Interacting residues definition

We labeled the interacting residues based on SASA and atom distance. Naccess V2.1.1 and Bio.PDB module were employed to calculate SASA values for each residue in antibody and antigen, respectively. The residues with a SASA loss ($\Delta$SASA) in binding of more than 1 $Å^2$ were classified as interacting residues. In addition, we also defined another set of interacting paratope-epitope residues by a distance cutoff of 6 Å. Two amino acids are considered interacting residues if they have at least one pair of non-hydrogen atoms within a distance of 6 Å.

## Data integration and website implementation

The main data processing algorithm is implemented in Python. The front-end web interface of AACDB was constructed by HTML and enhanced with JavaScript, CSS, and Bootstrap technologies. We developed a dynamic 3D structure visualization window based on PV, a WebGL-based protein viewer, inspired by *Dunbar et al., 2014*. All the data were managed within the MySQL database system. For the back-end functionality, PHP is utilized to enable data browsing, searching, and downloading features.

## Acknowledgements

We are grateful to Steve's Scholar Memory Team for polishing our final manuscript. This work was supported by the National Natural Science Foundation of China [grant numbers: 62071099, 62371112] and Sichuan Provincial Science and Technology Support Program [grant numbers: 2024NSFSC0636].

## Additional information

### Funding

| Funder | Grant reference number | Author |
|---|---|---|
| National Natural Science Foundation of China | 62071099 | Jian Huang |
| National Natural Science Foundation of China | 62371112 | Jian Huang |
| Sichuan Provincial Science and Technology Support Program | 2024NSFSC0636 | Jian Huang |

The funders had no role in study design, data collection and interpretation, or the decision to submit the work for publication.

### Author contributions
Yuwei Zhou, Conceptualization, Software, Writing – original draft; Wenwen Liu, Validation, Visualization; Ziru Huang, Validation; Yushu Gou, Siqi Liu, Lixu Jiang, Yue Yang, Data curation; Jian Huang, Supervision, Funding acquisition

### Author ORCIDs
Yuwei Zhou ⓘ https://orcid.org/0009-0006-7787-1711
Wenwen Liu ⓘ https://orcid.org/0009-0007-2591-9856
Ziru Huang ⓘ https://orcid.org/0009-0006-2380-3676
Yushu Gou ⓘ https://orcid.org/0009-0008-4613-3024
Siqi Liu ⓘ https://orcid.org/0009-0008-4918-5903
Lixu Jiang ⓘ https://orcid.org/0000-0001-6800-7063
Yue Yang ⓘ https://orcid.org/0009-0002-9935-0133
Jian Huang ⓘ https://orcid.org/0000-0003-3282-8892

Reviewer #1 (Public review): https://doi.org/10.7554/eLife.104934.3.sa1
Reviewer #2 (Public review): https://doi.org/10.7554/eLife.104934.3.sa2
Author response https://doi.org/10.7554/eLife.104934.3.sa3

## Additional files

### Supplementary files
MDAR checklist

### Data availability
All processed antigen-antibody complex data in AACDB are freely accessible at http://i.uestc.edu.cn/AACDB without registration requirements.

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
