## [Editor Report · eLife Assessment]

This **useful** manuscript provides a newly curated database (termed AACDB) of antibody-antigens structural information, alongside annotations that are either taken and from the PDB, or added de-novo. Sequences, structures, and annotations can be easily downloaded from the AACDB website, speeding up the development of structure-based algorithms and analysis pipelines to characterize antibody-antigen interactions. The methodology presented for this data curation is **solid**. The curated dataset will be of broad interest and value to researchers interested in antibody-antigen interactions.

---

## [Referee Report · Reviewer #1 (Public review)]

This work introduces and describes a useful curation pipeline of antibody-antigen structures downloaded from the PDB database. The antibody-antigen structures are presented in a new database called AACDB - with associated website - alongside annotations that were either corrected from those present in the PDB database, or added de-novo with solid methodology. Sequences, structures and annotations can be very easily downloaded from the AACDB website, speeding up the development of structure-based algorithms and analysis pipelines to characterize antibody-antigen interactions. However, AACDB is missing some important annotations that I believe would greatly enhance its usefulness, such as binding affinity annotations.

I think the potentially most significant contribution of this database is the manual data curation to fix errors present in the PDB entries, by cross-referencing with the literature. The authors also seem to describe, whenever possible, the procedures they took to correct the annotations.

I have personally verified some of the examples presented by the authors, and found that SAbDab appears to fix the mistakes related to mis-identification of antibody chains, but not other annotations.

"(1) the species of the antibody in 7WRL was incorrectly labeled as "SARS coronavirus B012" in both PDB and SabDab" → I have verified the mistake and fix, and that SAbDab does not fix is, just uses the pdb annotation.

"(2) 1NSN, the resolution should be 2.9 , but it was incorrectly labeled as 2.8" → I have verified the mistake and fix, and that saabdab does not fix it, just uses the PDB annotation.

"(3) mislabeling of antibody chains as other proteins (e.g. in 3KS0, the light chain of B2B4 antibody was misnamed as heme domain of flavocytochrome b2)" → SAbDab fixes this as well in this case.

"(4) misidentification of heavy chains as light chains (e.g. both two chains of antibody were labeled as light chain in 5EBW)" → SAbDab fixes this as well in this case.

I believe the splitting of the pdb files is a valuable contribution as it standardizes the distribution of antibody-antigen complexes. Indeed, there is great heterogeneity in how many copies of the same structure are present in the structure uploaded to the PDB, generating potential artifacts for machine learning applications to pick up on. That being said, I have two thoughts both for the authors and the broader community. First, in the case of multiple antibodies binding to different epitopes on the same antigen, one should not ignore the potentially stabilizing effect that the binding of one antibody has on the complex, thereby enabling the binding of the second antibody. In general, I urge the community to think about what is the most appropriate spatial context to consider when modeling the stability of interactions from crystal structure data. Second, and in a similar vein, some antigens occur naturally as homomultimers - e.g. influenza hemagglutinin is a homotrimer. Therefore, to analyze the stability of a full-antigen-antibody structure, I believe it would be necessary to consider the full homo-trimer, whereas in the current curation of AACDB with the proposed data splitting, only the monomers are present.

I think the annotation of interface residues is a very useful addition to structural datasets.

I am, however, not convinced of the utility of *change* in SASA as a useful metric for identifying interacting residues, beyond what is already identified via pairwise distances between the antibody and antigen residues. If we had access to the unbound conformation of most antibodies and antigens, then we could analyze the differences in structural conformations upon binding, which can be in part quantified by change in SASA. However, as only bound structures are usually available, one is usually force to approximate a protein's unbound structure by computationally removing its binding partner - as it seems to me the authors of this work are doing.

Some obvious limitations of AACDB in its current form include:

AACDB only contains entries with protein-based antigens of at most 50 amino-acids in length. This excludes non-protein-based antigens, such as carbohydrate- and nucleotide-based, as well as short peptide antigens https://www.biorxiv.org/content/10.1101/2023.12.10.570461v1.

AACDB does not include annotations of binding affinity, which are present in SAbDab and have been proven useful both for characterizing drivers of antibody-antigen interactions (cite https://www.sciencedirect.com/science/article/pii/S0969212624004362?via%3Dihub) and for benchmarking antigen-specific antibody-design algorithms cite.

---

## [Referee Report · Reviewer #2 (Public review)]

Summary:

Antibodies, thanks to their high binding affinity and specificity to cognate protein targets, are increasingly used as research and therapeutic tools. In this work, Zhou et al. have created, curated and made publicly available a new database of antibody-antigen complexes to support research in the field of antibody modelling, development and engineering.

Strengths:

The authors have performed a manual curation of antibody-antigen complexes from the Protein Data Bank, rectifying annotation errors; they have added two methods to estimate paratope-epitope interfaces; they have produced a web interface capable of effective visualisation and of summarising the key useful information in one page. The database is also cross-linked to other databases that contain information relevant to antibody developability and therapeutic applications.

Weaknesses:

The database does not import all the experimental information from PDB and contains only complexes with large protein targets.

Comments on revisions: I thank the authors for having incorporated my feedback and I look forward to the next releases of this database.

---

## [Author Response]

The following is the authors’ response to the original reviews

**Public reviews:**

**Reviewer #1:**
(1) This manuscript introduces a useful curation pipeline of antibody-antigen structures downloaded from the PDB database. The antibody-antigen structures are presented in a new database called AACDB, alongside annotations that were either corrected from those present in the PDB database or added de-novo with a solid methodology. Sequences, structures, and annotations can be very easily downloaded from the AACDB website, speeding up the development of structure-based algorithms and analysis pipelines to characterize antibody-antigen interactions. However, AACDB is missing some key annotations that would greatly enhance its usefulness.Here are detailed comments regarding the three strengths above:I think potentially the most significant contribution of this database is the manual data curation to fix errors present in the PDB entries, by cross-referencing with the literature. However, as a reviewer, validating the extent and the impact of these corrections is hard, since the authors only provided a few anecdotal examples in their manuscript.I have personally verified some of the examples presented by the authors and found that SAbDab appears to fix the mistakes related to the misidentification of antibody chains, but not other annotations.(a) "the species of the antibody in 7WRL was incorrectly labeled as "SARS coronavirus B012" in both PDB and SabDab" → I have verified the mistake and fix, and that SAbDab does not fix is, just uses the pdb annotation.(b) "1NSN, the resolution should be 2.9 , but it was incorrectly labeled as 2.8" → I have verified the mistake and fix, and that sabdab does not fix it, just uses the PDB annotation.(c) "mislabeling of antibody chains as other proteins (e.g. in 3KS0, the light chain of B2B4 antibody was misnamed as heme domain of flavocytochrome b2)" → SAbDab fixes this as well in this case.(d) "misidentification of heavy chains as light chains (e.g. both two chains of antibody were labeled as light chain in 5EBW)" → SAbDab fixes this as well in this case.I personally believe the authors should make public the corrections made, and describe the procedures - if systematic - to identify and correct the mistakes. For example, what was the exact procedure (e.g. where were sequences found, how were the sequences aligned, etc.) to find mutations? Was the procedure run on every entry?

We appreciate the reviewer’s valuable feedback. Our correction procedures combined manual curation with systematic sequence analysis. While most metadata discrepancies were resolved through cross-referencing original literature, we implemented a structured approach for identifying mutations in specific cases. For PDB entries labeled as variants (e.g., "Bevacizumab mutant" or "Ipilimumab variant Ipi.106") where the "Mutation(s)" field was annotated as "NO," we retrieved the canonical therapeutic antibody sequence from Thera-SAbDab, then performed pairwise sequence alignment against the PDB entry using BLAST program to identified mutated residues.

This procedure was not applied to all entries, as mutations are context-dependent. Therapeutic antibodies have well-defined reference sequences, enabling systematic alignment. For antibodies lacking unambiguous wild-type references (e.g., research-grade or non-therapeutic antibodies), mutation annotations were directly inherited from the PDB or literature.

All corrections have been publicly archived in AACDB. We have added a detailed discussion of this issue in the section “2.3 Metadata” of revised manuscript.

(2) I believe the splitting of the pdb files is a valuable contribution as it standardizes the distribution of antibody-antigen complexes. Indeed, there is great heterogeneity in how many copies of the same structure are present in the structure uploaded to the PDB, generating potential artifacts for machine learning applications to pick up on. That being said, I have two thoughts both for the authors and the broader community. First, in the case of multiple antibodies binding to different epitopes on the same antigen, one should not ignore the potentially stabilizing effect that the binding of one antibody has on the complex, thereby enabling the binding of the second antibody. In general, I urge the community to think about what is the most appropriate spatial context to consider when modeling the stability of interactions from crystal structure data. Second, and in a similar vein, some antigens occur naturally as homomultimers - e.g. influenza hemagglutinin is a homotrimer. Therefore, to analyze the stability of a full-antigen-antibody structure, I believe it would be necessary to consider the full homo-trimer, whereas, in the current curation of AACDB with the proposed data splitting, only the monomers are present.

We sincerely appreciate the reviewer’s insightful comments regarding the splitting of PDB files and we appreciate the opportunity to address the reviewer’s thoughtful concerns.

Firstly, when two antibodies bind to distinct epitopes on the same antigen, we would like to clarify that this scenario can be divided into two cases based on the experimental context: Case1: When two antibodies bind to distinct epitopes on the same antigen, and their complexes are determined in separate structures. For example, SAR650984 (PDB: 4CMH) and daratumumab (PDB: 7DHA) target CD38 at non-overlapping epitopes. These two antibody-antigen complexes were determined independently, and their structures do not influence each other. Case 2 : When the crystal structure contains a ternary complex with two antibodies and an antigen, as in the example of 6OGE discussed in Section 2.2 of our manuscript. After reviewing the original literature, the experiment confirmed that the order of Fab binding does not affect the formation of the ternary complex, and the binding of one antibody does not enhance the binding of the other. This supports the rationale for splitting 6OGE into two separate structures. However, we acknowledge that not all ternary complexes in the PDB provide such detailed experimental descriptions in their original literature. We agree with the reviewer that in some cases, one antibody may stabilize the structure to facilitate the binding of a second antibody. For instance, in 3QUM, the 5D5A5 antibody stabilizes the structure, enabling the binding of the 5D3D11 antibody to human prostate-specific antigen. Such sandwich complexes are indeed valuable for identifying true epitopes and paratopes. Importantly, splitting the structure does not alter the interaction sites.

Secondly, we fully agree with the reviewer that for antigens that naturally exist as homomultimers (e.g., influenza hemagglutinin as a homotrimer), the full multimeric structure should be considered when analyzing stability. In such cases, users can directly utilize the original PDB structures provided in their multimeric form. Our splitting approach is intended to provide an additional option for cases where monomeric analysis is sufficient or preferred, but it does not preclude the use of the original multimeric structures when necessary.

(3) I think the manuscript is lacking in justification about the numbers used as cutoffs (1A^2 for change in SASA and 5A for maximum distance for contact) The authors just cite other papers applying these two types of cutoffs, but the underlying physico-chemical reasons are not explicit even in these papers. I think that, if the authors want AACDB to be used globally for benchmarks, they should provide direct sources of explanations of the cutoffs used, or provide multiple cutoffs. Indeed, different cutoffs are often used (e.g. ATOM3D uses 6A instead of 5A to determine contact between a protein and a small molecule https://datasets-benchmarks-proceedings.neurips.cc/paper/2021/hash/c45147dee729311ef5b5c3003946c48f-Abstract-round1.html). I think the authors should provide a figure with statistics pertaining to the interface atoms. I think showing any distribution differences between interface atoms determined according to either strategy (number of atoms, correlation between change in SASA and distance...) would be fundamental to understanding the two strategies. I think other statistics would constitute an enhancement as well (e.g. proportion of heavy vs. light chain residues).Some obvious limitations of AACDB in its current form include:AACDB only contains entries with protein-based antigens of at most 50 amino acids in length. This excludes non-protein-based antigens, such as carbohydrate- and nucleotide-based, as well as short peptide antigens.AACDB does not include annotations of binding affinity, which are present in SAbDab and have been proven useful both for characterizing drivers of antibody-antigen interactions (cite https://www.sciencedirect.com/science/article/pii/S0969212624004362?via%3Dihub) and for benchmarking antigen-specific antibody-design algorithms (cite https://www.biorxiv.org/content/10.1101/2023.12.10.570461v1).

We thank the reviewer for raising this critical point about the cutoff values used in AACDB. In the current study, the selection of the threshold value is very objective; the threshold chosen in the manuscript is summarized based on existing literature, and we have provided more literature support in the manuscript. The criteria for defining interacting amino acids in established tools, typically do not set the ΔSASA exceed 1 Å2 and the distance exceed 6 Å. While our manuscript emphasizes widely accepted thresholds for consistency with prior benchmarks, AACDB explicitly provides raw ΔSASA and distance values for all annotated residues. Users can dynamically filter the data from downloaded files by excluding entries exceeding their preferred thresholds (e.g., selecting 5Å instead of 6Å). This ensures adaptability to diverse research needs. In the revised version, we reset the distance threshold to 6 Å and calculated the interacting amino acids in order to give the user a wider range of choices. In the section “3.2 Database browse and search” of revised manuscript, we provide a description of the flexible choice of thresholds for practical use.

Furthermore, distance and ΔSASA are two distinct metrics for evaluating interactions. Distance directly quantifies spatial proximity between atoms, reflecting physical contacts such as van der Waals interactions or hydrogen bonds, and is ideal for identifying direct spatial adjacency. ΔSASA, on the other hand, measures changes in solvent accessibility of residues during binding, capturing the contribution of buried surfaces to binding free energy. Even for residues not in direct contact, reduced SASA due to conformational changes may indicate indirect functional roles.

As demonstrated through comparisons on the detailed information pages, the sets of interacting amino acids defined by these two methods differ by only a few residues, with no significant variation in their overall distributions. However, since interaction patterns vary significantly across different complexes, analyzing residue distributions across all structures using both criteria is not feasible.

We thank the reviewer for highlighting these limitations. AACDB currently focuses on protein-based antigens ≤50 amino acids to prioritize structural consistency, which excludes non-protein antigens and shorter peptides. While affinity annotations are critical for benchmarking antibody design tools, these data were not integrated in this release due to insufficient data verification caused by internal team constraints. We acknowledge these gaps and plan to expand antigen diversity and incorporate affinity metrics in future updates.

**Reviewer #2:**
Summary:Antibodies, thanks to their high binding affinity and specificity to cognate protein targets, are increasingly used as research and therapeutic tools. In this work, Zhou et al. have created, curated, and made publicly available a new database of antibody-antigen complexes to support research in the field of antibody modelling, development, and engineering.Strengths:The authors have performed a manual curation of antibody-antigen complexes from the Protein Data Bank, rectifying annotation errors; they have added two methods to estimate paratope-epitope interfaces; they have produced a web interface that is capable of both effective visualisation and of summarising the key useful information in one page. The database is also cross-linked to other databases that contain information relevant to antibody developability and therapeutic applications.Weaknesses:The database does not import all the experimental information from PDB and contains only complexes with large protein targets.

Thank you for the valuable feedback. As previously responded to Reviewer 1, due to limitations within our team, comprehensive data integration from PDB has not been achieved in the current version. We acknowledge the significance of expanding the database to encompass a broader range of experimental information and complexes with diverse target sizes. Regrettably, immediate updates to address these limitations are not feasible at this time. Nevertheless, we are committed to enhancing the database in upcoming upgrades to provide users with a more comprehensive and inclusive resource

**Recommendations for the authors:**

**Reviewer #1:**
(1) Line 194: "produce" → "produced"

We thank the reviewer for the feedback. We have checked the grammar and spelling carefully in the revised manuscript.

(2) As mentioned in the public review, I think adding binding affinity annotations would greatly enhance the use cases for the database.

We thank the reviewer for the suggestion. As the response in “Public review”. Due to team constraints, these data are not integrated into this release but are being collated. We recognize these gaps and plan to expand antigenic diversity and incorporate affinity metrics in future updates.

(3) I think adding a visualization of interface atoms and contacts on an entry's webpage would be useful for someone exploring specific entries. It also would be useful if the authors provided a pymol command to select interface residues since that's a procedure any structural biologist is likely to do.

We sincerely appreciate the reviewer’s constructive suggestions. In response to the request for enhanced visualization and accessibility of interface residue information, we have implemented the following improvements: (1) Web Interface Visualization. On the entry-specific webpage, we have added an interactive visualization window that highlights the antigen-antibody interaction interface using distinct colors. The interaction interface visualization has been incorporated into Figure 5 of the revised manuscript, with a detailed description. (2) PyMOL Command Accessibility. The “Help” page now provides step-by-step PyMOL commands to select and visualize interface residues.

(4) I think the authors should provide headers to the files containing interface residues according to the change-in-SASA criterion, as they do for those computed according to contact. This would avoid unnecessary confusion - however slight - and make parsing easier. I was initially confused by the meaning of the last column, though after a minute I understood it to be the change in SASA.

We thank the reviewer for providing such detailed feedback. We thank the reviewer for the comment and the suggestion. We have provided headers for the files of the interacting residues defined by ΔSASA.

(5) Line 233: "AACDB's data processing pipeline supports mmCIF files" → The meaning and implications of this statement are not obvious to me, and are mentioned nowhere else in the paper. Do you mean that in AACDB there are structure entries that the RCSB PDB database only has in mmCIF file format, and not .pdb format? So, effectively, there are some entries in AACDB that are not in any other antibody-specific database?I checked and, as of Dec 3rd, 2024, there are 41 structures in AACDB that are NOT in SAbDab. Manually checking 5 of those 41 structures, none are mmCIF-only structures.

We thank the reviewer for the valuable comment. Because of the size of the structures within certain entries, representing them in a single PDB format data file is not feasible due to the excessive number of atoms and polymer chains they contain. As a result, PDB stores these structures in “mmcif” format files. In AACDB, 47 entries, such as 7SOF, 7NKT, 7B27, and 6T9D, are only available in the “mmCIF” format from the PDB. The “.pdb” and “.cif” files contain atomic coordinates in distinct text formats, and the segmentation of these structure files is automatically conducted based on manually annotated antibody-antigen chains. To accommodate this, we have incorporated these considerations into our file processing pipeline, thereby enabling a fully automated file segmentation process. Additionally, we employed Naccess to calculate interatomic distances. However, since this software only accepts .pdb format files as input, we also converted all split .cif files into .pdb format within our fully automated pipeline. We apologize for the lack of clarity in the original manuscript and have included a more detailed explanation in the "2.2 PDB Splitting" section of the revised manuscript.

**Reviewer #2:**
(1) In SabDab and PDB, experimental binding affinities are also reported: could the authors comment on whether they also imported this information and double-checked it against the original paper? If it wasn't imported, that might discourage some users and should be considered as an extension for the future.

We thank the reviewer for the comment and the suggestion. As the response in “Public review”. Due to current resource constraints, quantitative affinity data has not been incorporated into this release but is undergoing systematic curation. We explicitly recognize these limitations and propose a two-pronged strategy for future iterations: (1) broadening antigen diversity coverage through expanded structural sampling, and (2) integrating quantitative binding affinity measurements. In the Discussion section, we have included description outlining the planned enhancements.

(2) Line 49-50: the references mentioned in connection to deep learning methods for antibody-antigen predictions seem a bit limited given the amount of articles in this field, with 3 of 4 references on one method only (SEPPA), could the authors expand this list to reflect a bit more the state of the art?

We thank the reviewer for the suggestion. We agree that more relevant studies should be listed and therefore more references are provided in the revised manuscript.

When mentioning the limitations of the existing databases, it feels a bit that the criticism is not fully justified. For instance:Line 52-53: could the authors elaborate on the reasons why such an identification is challenging? (Isn't it possible to make an efficient database-filtered search? Or rather, should one highlight that a more focussed resource is convenient and why?)

Thank you for feedback. In this study, the keywords "antibody complex," "antigen complex," and "immunoglobulin complex," were employed during data collection. PDB returned over 30,000 results, of which only one-tenth met our criteria after rigorous filtering. This demonstrates that keyword searches, while useful, inherently limit result precision and introduce substantial redundancy, likely due to the PDB's search mechanism. That’s why we illustrated the significant challenges in identifying antibody-antigen complexes from general protein structures in the PDB.

Line 55: reading the website http://www.abybank.org/abdb/, it would be fairer to say that the web interface lacks updates, as the database and the code have gone through some updates. Could the authors provide a concrete example of the reason why: 'The AbDb database currently lacks proper organization and management of this valuable data.'?

We thank the reviewer for highlighting this issue. In our original manuscript, the statement that the AbDb database "lacks proper organization and management" was based on the absence of explicit statement regarding data updates on its official website at the time of submission, even though internal updates to its content may have occurred. We fully respect the long-standing contributions of AbDb to antibody structural research, and our comments were solely directed at the specific state of the database at that time. As the reviewer noted, following the release of our preprint, we have also taken note of AbDb's recent updates. To reflect the latest developments and avoid potential misinterpretation, we have revised the original statement in revised manuscript.

Also 'this rapid updating process may inadvertently overlook a significant amount of information that requires thorough verification,': it's difficult for me to understand what this means in practice. Could the authors clarify if they simply mean that SabDab collects information from PDB and therefore tends to propagate annotation errors from there? If yes, I think it's enough to state it in these terms, and for sure I agree that the reason is that correcting these annotation errors requires a substantial amount of work.

We thank the reviewer for providing such detailed feedback on the manuscript. We acknowledge that SabDab represents a highly valuable contribution to the field, and its rapid update mechanism has significantly advanced related research areas. However, as stated by the reviewer, we aim to clarify that SabDab primarily relies on automated metadata extraction from the PDB for annotation, and its rapid update process inherently inherits raw data from upstream sources. According to their paper, manual curation is only applied when the automated pipeline fails to resolve structural ambiguities. This workflow—dependent on PDB annotations with limited manual verification—may propagate errors provided by PDB. Examples include species misannotation and mutation status misinterpretation. We fully agree with the reviewer's observation that correcting errors in such cases necessitates labor-intensive manual curation, which is a core motivation for our study.

Line 86: why 'Structures that consisted solely of one type of antibody were excluded'? Why exclude complexes with antigens shorter than 50 amino acids? These complexes are genuine antibody-antigen complexes.

We thank the reviewer for the valuable question. The AACBD database is dedicated to curating structural data of antigen-antibody complexes. Structures featuring only a single antibody type are classified as free antibodies and systematically excluded from the database due to the absence of protein-bound partners. During data screening , we retained sequences shorter than 50 amino acids by categorizing them as peptides rather than eliminating them outright. The current release exclusively encompasses complexes with protein-based antigens. Meanwhile, complexes involving peptide, haptens, and nucleic acid antigens are undergoing systematic curation, with planned inclusion in future updates to broaden antigen category representation.

Line 96 needs a capital letter at the beginning.Line 107: 'this would generate' → 'this generates' (given it is something that has been implemented, correct?).Line 124: missing an 'of'.Line 163: inspiring by -> inspired by.

Thank you for feedback. All of the above grammatical or spelling errors have been revised in the manuscript.

Line 109-111: apart from the example, it would be good to spell out the general rule applied to anti-idiotypic antibodies.

We thank the reviewer for the valuable feedback. For anti-idiotypic antibodies complex. the partner antibody is treated as a dual-chain antigen, , necessitating individual evaluation of heavy chain and light chain interactions with the anti-idiotypic component. We have given a general rule for anti-idiotypic antibodies in section “2.2 PDB splitting” of revised manuscript.

Line 155-159: could the authors provide references for the two choices (based on sasa and any-atom distance) that they adopted to define interacting residues?

We thank the reviewer for the comment and the suggestion. As the same as the response to reviewer #1 in Public review. The interacting residues definition and the threshold chosen in the manuscript is summarized based on existing literature. We have added additional references for support in section “1.Introduction”. Our resource does not provide a fixed amino acid list. Instead, all interacting residues are explicitly documented alongside their corresponding ΔSASA (solvent-accessible surface area changes) and intermolecular distances, allowing researchers to flexibly select residue pairs based on customized thresholds from downloadable datasets. Furthermore, aligning with widely adopted criteria in current literature—where interactions are defined by ΔSASA >1 Å² and atomic distances <6 Å, we have recalibrated our analysis in the revised version. Specifically, we replaced the previous 5 Å distance threshold with a 6 Å cutoff to recalculate interacting residues.

Line 176-178: could the authors re-phrase this sentence to clarify what they mean by 'change in the distribution'?

We thank the reviewer for the suggestion. Our search was conducted with an end date of November 2023. However, Figure 3B includes an entry dated 2024. Upon reviewing this record, we identified that the discrepancy arises from the supersession of the 7SIX database entry (originally released in December 2022) by the 8TM1 version in January 2024. This version update explains the apparent chronological inconsistency. We regret any lack of clarity in our original description and have revised the corresponding section in the manuscript to explicitly clarify this change of database.

Caption Figure 3: please spell out all the acronyms in the figure. Provide the date when the last search was performed (i.e., the date of the last update of these statistics).

We thank the reviewer for the comment. We have systematically expanded all acronyms and included update dates for statistics in the legend of Figure 3. Corresponding changes have also been made to the statistical pages on the website.

Finally, it would be advisable to do a general check on the use of the English language (e.g. I noted a few missing articles). In Figure 5 DrugBank contains typos.

We sincerely appreciate the reviewer's meticulous attention to linguistic precision. We have corrected the typographical error in Figure 5 and conducted a comprehensive review of the entire manuscript to ensure accuracy and clarity.